# Effects of Inoculating Autochthonous Starter Cultures on Changes of *N-*Nitrosamines and Their Precursors in Chinese Traditional Fermented Fish during In Vitro Human Digestion

**DOI:** 10.3390/foods13132021

**Published:** 2024-06-26

**Authors:** Han Li, Qian Li, Qi Wang, Jiwang Chen, Wenshui Xia, E Liao

**Affiliations:** 1College of Food Science and Engineering, Wuhan Polytechnic University, Wuhan 430023, China; 2Hubei Key Laboratory for Processing and Transformation of Agricultural Products, Ministry of Education, Wuhan 430023, China; 3National R&D Center for Se-Rich Agricultural Products Processing Technology, Wuhan 430023, China; 4School of Food Science and Technology, Jiangnan University, Wuxi 214122, China

**Keywords:** Chinese traditional fermented fish, biogenic amines, nitrite, *N-*nitrosamines, in vitro human digestion

## Abstract

The objective of this research was to investigate the impact of inoculating autochthonous starter cultures on the alterations in microorganisms, biogenic amines, nitrite, and *N-*nitrosamines in Chinese traditional fermented fish products (CTFPs) during in vitro human digestion. The results revealed that gastric digestion significantly (*p* < 0.05) inhibited the proliferation of lactic acid bacteria, yeast, *Staphylococcus*, and Enterobacteriaceae, whereas various microorganisms proliferated extensively during small intestine digestion. Meanwhile, small intestine digestion could significantly increase (*p* < 0.05) levels of putrescine, cadaverine, and tyramine. The reduced content observed in inoculated fermentation groups suggests that starter cultures may have the ability to deplete biogenic amines in this digestion stage. Gastric digestion significantly (*p* < 0.05) inhibited nitrite accumulation in all CTFPs samples. Conversely, the nitrite content increased significantly (*p* < 0.05) in all groups during subsequent small intestine digestion. However, the rise in the inoculated fermentation groups was smaller than that observed in the spontaneous fermentation group, indicating a potentially positive role of inoculated fermentation in inhibiting nitrite accumulation during this phase. Additionally, gastric digestion significantly (*p* < 0.05) elevated the levels of *N-*nitrosodimethylamine (NDMA) and *N-*nitrosopiperidine in CTFPs. Inoculation with *L. plantarum* 120, *S. cerevisiae* 2018, and mixed starter cultures (*L. plantarum* 120, *S. cerevisiae* 2018, and *S. xylosus* 135 [1:1:1]) effectively increased the degree of depletion of NDMA during this digestion process.

## 1. Introduction

*N-*nitroso-compounds, organic compounds containing nitroso (-NO) functional groups, are further divided into *N-*nitrosamines (NAs) and *N*-nitrosamides based on their structures. NAs, widely regarded as potent carcinogens worldwide, are commonly found in drinking water, tobacco smoke, food, and household products [1]. Concerning food source contamination, the origin of NAs is frequently linked to the processing of meat products [2,3]. Although raw meat contains low levels of naturally occurring amines, the decomposition of proteins into polypeptides and amino acids during processing, such as curing and fermentation, contributes significantly to the production of biogenic amines (BAs) [4]. Furthermore, nitrite, commonly added during processing to enhance the color, flavor, and quality of cured meats, can react with BAs as one of the precursor substances, thereby increasing the risk of excessive NAs in processed meat products [5].

Fermentation, recognized as one of the oldest preservation methods, is a valuable asset in human history [6]. Widely utilized in the processing of perishable meat and fish products, fermentation enhances shelf life and imparts sensory attributes [7,8,9]. Chinese traditional fermented fish products (CTFPs) are solid fermented fish products popular in southern China because of their low fishy taste, distinct flavor, and easily absorbed nutrition [10]. However, our previous survey conducted on CTFPs in the Chinese market reported that 63.6% of fermented fish products exceeded the limit (10 μg/kg) suggested by the United States Department of Agriculture for volatile NAs [11], indicating a significant contamination issue in these products. In recent years, biological fermentation technology has emerged as an effective approach to mitigate the accumulation of NAs in traditional fermented products. Sun et al. [12] demonstrated a high inhibitory effect on *N-*nitrosodiethylamine (NDEA), *N-*nitrosodipropylamine (NDPA), *N-*nitrosodiphenylamine (NDPheA), and *N-*nitrosopiperidine (NPIP) in Harbin dry sausages through inoculation with *Lactobacillus curvatus*. Shao et al. [13] similarly found that *Lactobacillus pentosus* R3 is a reliable way to reduce the content of *N-*nitrosodimethylamine (NDMA) in fermented sausages, especially after the ripening and cooking steps.

NAs accumulate in humans not only through the direct ingestion of contaminated food but also via nitrosation reactions involving BAs and nitrite in the acidic stomach environment during digestion [14]. Approximately 45–75% of the identified NAs are reported to be formed endogenously [15]. Mirvish [16] further confirmed that the acid-catalyzed formation of NAs in the stomach is primarily responsible for initiating gastric cancer. It is noteworthy that dietary intake of BAs and nitrites may potentially increase NAs (*N*-nitrosamines) exposure within the body. Therefore, in addition to controlling the levels of NAs and their precursors in products, gaining a better understanding of the endogenous synthesis of NAs is crucial for enhancing food safety. The utilization of in vitro digestion provides a means of elucidating the changes in NAs and their precursors within the food matrix for endogenous exposure in the human body. Latest advances have conducted the changes in NAs during in vitro human digestion on Korean kimchi [17], pork patties [18], and sausages [19]. However, the life-threatening status of compounds such as BAs, nitrite, and NAs endogenous formation associated with the intake of fermented aquatic products has received limited attention. Moreover, there remains a dearth of research regarding the impact of inoculating autochthonous starter cultures on the fate of NAs and their precursors in CTFPs during digestion.

This study aimed to investigate the impact of inoculating autochthonous starter cultures, including *Lactobacillus plantarum* 120, *Saccharomyces cerevisiae* 2018, and *Staphylococcus xylosus* 135, on the alterations of microorganisms, nitrite, BAs, and NAs in CTFPs during in vitro human digestion. These experimental results were expected to provide new insights into the dynamic changes of NAs and their precursors in fermented freshwater fish products during different phases of digestion.

## 2. Materials and Methods

### 2.1. Starter Cultures

The starter cultures chosen, based on their technological and antimicrobial properties, were strains of *L. plantarum* 120, *S. cerevisiae* 2018, and *S. xylosus* 135, previously isolated and identified from Suanyu, a type of Chinese traditional fermented fish [20]. *L. plantarum* 120 underwent subculture twice in Man Rogosa and Sharpe (MRS) broth at 37 °C under static conditions for 48 h. *S. cerevisiae* 2018 underwent subculture twice in Yeast Extract Peptone Dextrose (YPD) broth at 37 °C under static conditions for 24 h. *S. xylosus* 135 underwent subculture twice in Mannitol Salt Agar (MSA) broth at 37 °C with shaking at 150 rpm for 48 h. After incubation, all cell pellets were harvested using a high-speed cryogenic centrifuge (Model 4K15, Sigma Laborzentrifugen, Osterode, Germany) at 10,000× *g* for 15 min at 4 °C. Post harvest, pellets underwent two washes with saline water (0.9% NaCl, *w*/*v*), followed by re-suspension in 10 mL of saline water. Subsequently, the number of bacterial cells in each suspension was adjusted to 7–8 log cfu/mL by measuring the optical density at 600 nm with a spectrophotometer (UV 1000, Techcomp Scientific Instruments Co., Ltd., Shanghai, China).

### 2.2. CTFPs Preparation

Following the method outlined by Zang et al. [21], the CTFPs were prepared, with each group totaling approximately 500 g in weight. The natural fermentation group, without any starter cultures, was defined as NS. Based on their abilities to degrade BAs [22], the prepared starter cultures *L. plantarum* 120, *S. cerevisiae* 2018, and *S. xylosus* 135 were used individually to inoculate samples (1%, *v*/*w*), designated as the LP-120, SC-2018, and SX-135 groups, respectively. In another group, a mixed starter culture (*L. plantarum* 120, *S. cerevisiae* 2018, and *S. xylosus* 135 [1:1:1]) was inoculated with equal amounts of the individual starters mentioned above, designated as the MS group. These products were placed in tightly sealed glass jars with water sealing, and fermentation occurred at 24 °C until the pH of the CTFPs reached 4.0–4.5, which is considered safe for consumption [20]. Every week, approximately 100 g samples were randomly collected from each group, vacuum-packed using an MS1160 vacuum sealer (MagicSeal Ltd., Christchurch, New Zealand), and stored at −50 °C (MDF-U53V, SANYO Electric Co., Ltd, Osaka, Japan) for subsequent analysis within two weeks. Furthermore, protease activities were determined immediately to mitigate the effects of the freezing and thawing process.

### 2.3. In Vitro Human Digestion System

Based on the methods outlined by Kim and Hur [23] and Grassi et al. [24] with minor adjustments, an in vitro simulated digestion system was established. The flow diagram illustrating the process of the in vitro simulated digestion is depicted in Figure 1. A 5 g portion of grated CTFPs sample was thoroughly mixed with 10 mL of salt solution (140 mmol/L NaCl and 5 mol/L KCl) and incubated at 37 °C with shaking at 200 rpm for 10 min to simulate the oral digestion process. As the fermented fish samples contained negligible amounts of starch, amylase was not added to the simulation system. Following this, the pH of the oral digestive solution was adjusted to 2.0 using 1 mol/L hydrochloric acid. Then, 0.5 mL of pepsin solution (200 mg of pepsin dissolved in 5 mL of 0.1 mol/L hydrochloric acid) was added to attain a final pepsin concentration of 2000 U/mL in the digestive solution. The mixture was then incubated at 37 °C with shaking at 200 rpm for 1 h to simulate the digestive process of gastric juice. Subsequently, the pH of the digestive solution was adjusted to 6.5 with 1 mol/L sodium bicarbonate solution. Next, 2.5 mL of a trypsin-bile salt-lipase mixture (comprising 75 mg pancreatin, 450 mg porcine bile salt, and 25 mg lipase dissolved in 37.5 mL of 0.1 mol/L sodium bicarbonate) was introduced into the digestive solution. The mixture was then incubated at 37 °C with shaking at 200 rpm for 2 h to simulate the digestion process in the small intestine. Following digestion, samples were collected at various stages, and the digested mixtures were centrifuged at 12,000× *g* at 4 °C for 15 min. A portion of the pellet was immediately collected for microbiological testing, and the remaining pellet was stored at −50 °C for further analysis. The composition of the simulated saliva, gastric, and duodenal juices are listed in Table 1.

### 2.4. Microbiological Analysis

Duplicate samples (5 g each) were aseptically combined with 45 mL of normal saline (0.9% NaCl, *w*/*v*) in plastic pouches and homogenized at 25 °C for 120 s using a homogenizer (BM-400P, Truelab Laboratory Equipment Co., Ltd., Shanghai, China). Subsequently, a series of 10-fold dilutions of the homogenate were prepared for microbiological analysis. For each dilution (0.1 mL), inoculation was performed into the respective growth media to determine microbial counts. Lactic acid bacteria (LAB) were cultivated on MRS agar under anaerobic conditions at 37 °C for 48 h. Yeast was cultured on YPD agar under anaerobic conditions at 37 °C for 24 h. Staphylococcus was grown on MSA plates at 37 °C for 72 h. Enterobacteriaceae was enumerated on Violet Red Bile (VRB) agar and cultured at 37 °C for 24 h. The colony number was quantified as colony-forming units per gram (log cfu/g).

### 2.5. BAs Determination

BAs in fish samples were assessed following the method outlined by Qin et al. [25] with minor adjustments. Specifically, 5 g of samples were homogenized with 15 mL of 0.6 M HClO_4_ solution. The homogenate was centrifuged at 10,000× *g* for 15 min at 4 °C centrifuge (4K15, Sigma Laborzentrifugen, Osterode, Germany). These procedures were repeated twice, and the combined supernatants were adjusted to 50 mL using 0.6 M HClO_4_. Following this, 1 mL of the diluted supernatants was thoroughly mixed with 200 μL of 2 M NaOH, 300 μL of saturated NaHCO_3_, and 1 mL of 1% (*w*/*v*) dansyl-chloride (Sigma-Aldrich, Darmstadt, Germany) solution for derivatization. The mixture was then incubated at 40 °C in the dark for 45 min, and the reaction was halted by adding 100 μL of ammonia. After allowing it to stand at room temperature for 30 min, the final volume was diluted to 5 mL with acetonitrile. This diluted solution was centrifuged at 3000× *g* for 10 min at 4 °C using a centrifuge. Subsequently, the supernatant was filtered through a 0.22 μm membrane filter (Sartorius NY, Goettingen, Germany). Finally, 200 μL of the filtrate was transferred into a glass vial for subsequent analysis.

An RP-HPLC column (C18-Diamondsil, 25 cm × 4.6 mm, 5 μm) at a column temperature of 30 °C and a flow rate of 0.8 mL/min was utilized. The elution solvents consisted of ammonium acetate (0.1 M) for washing and acetonitrile for elution. A gradient elution process was employed: starting from 50% to 90% elution solvent over 0–35 min, then returning from 90% to 50% over 35–45 min. BAs were quantified via an ultraviolet detector set at 254 nm. Standard concentrations of BAs substances (1 mg/L, 5 mg/L, 10 mg/L, 25 mg/L, 50 mg/L, and 100 mg/L), including putrescine (PUT), cadaverine (CAD), tyramine (TYR), spermidine (SPD), and spermine (SPM), were analyzed under the same chromatographic conditions. Peak identification and quantification were conducted by comparing retention times and peak areas with BAs (biogenic amines) standards.

### 2.6. Nitrite Determination

For nitrite concentration determination, colorimetric nitrite assay based on the Griess reaction [26] was employed. Initially, 10 g of ground sample was mixed with 12.5 mL of saturated sodium borate solution, deproteinated, and defatted by the addition of 5 mL of 106 g/L potassium ferrocyanide solution and 5 mL of 220 g/L zinc acetate solution, followed by filtration. The resulting filtrate (40 mL) was transferred to a 50 mL colorimetric tube. Sequentially, 2 mL of sulfanilic acid (4 g/L) and 1 mL of N-1-naphtyethylene diamine dihydrochloride (2 g/L) were sequentially added. After dilution with distilled water to a final volume of 50 mL, the absorbances of the colored mixtures were measured at 538 nm using a spectrophotometer (UV 1000, Techcomp Scientific Instruments Co., Ltd., Shanghai, China) against the reagent blank.

### 2.7. NAs (N-Nitrosamines) Determination

CTFPs samples (10 g each) underwent extraction of NAs following the method described by Wu et al. [11] with minor adjustments. Each sample was mixed with 15 mL of acetonitrile in a 50 mL polypropylene tube and homogenized for 3 min using a vortex mixer (VORTEX 1, IKA, Staufen, Germany). After storage at −20 °C for 30 min, 4 g of MgSO_4_, 1 g of NaCl, and one stirring bar were introduced into the tube, followed by vigorous shaking for 30 s. Following centrifugation at 3800× *g* at 4 °C for 10 min (4K15, Sigma Laborzentrifugen, Osterode, Germany), 6 mL of supernatant was transferred to a tube containing 50 mg of primary secondary amine (PSA), 150 mg of octadecylsilane (C18E), and 900 mg of anhydrous Na_2_SO_4_. After shaking for 1 min and centrifuging at 3800× *g* at 4 °C for 10 min, 5 mL of supernatant was concentrated to 1 mL at room temperature using a concentrator (WD-12, Allsheng Instruments Co., Ltd., Hangzhou, China). Subsequently, the enriched sample was filtered through a 0.22 μm membrane filter (Sartorius NY, Goettingen, Germany) for GC-MS/MS analysis.

The detection and quantification of NAs were conducted using a GC-MS/MS system (TSQ Quantum XLS Ultra, Thermo Fisher Scientific Inc., Waltham, MA, USA). The injection volume was set at 10 μL. Chromatographic separation was achieved using a DB-WAX column (60 m × 0.25 mm × 0.25 μm) (Agilent Technologies Inc., Santa Clara, CA, USA). The injection port temperature was maintained at 250 °C. Initially, the column temperature was held at 40 °C for 3 min, then increased to 110 °C at a rate of 10 °C/min. Subsequently, it was further increased from 110 °C to 200 °C at a rate of 15 °C/min, and then from 200 °C to 240 °C at a rate of 40 °C/min. Helium served as the gas carrier at a flow rate of 25 mL/min. The transfer line temperature was kept constant at 250 °C. MS/MS analysis was conducted in multiple reaction monitoring (MRM) mode, with the ion source temperature set at 230 °C. Standard solutions of NAs, including NDMA, *N-*nitrosoethylmethylamine (NMEA), NPIP, *N-*nitrosopyrrolidine (NPYR), *N-*nitrosomorpholine (NMOR), and NDPheA, with concentrations ranging from 0.001 to 1.0 μg/mL were prepared by diluting with acetonitrile.

### 2.8. Statistical Analysis

The software SPSS version 25 (SPSS Inc., Chicago, IL, USA) was used to analyze all data. Data were expressed as the mean ± standard deviation from multiple groups of independent experiments. Differences among mean values were established using Duncan’s multiple range test, and significant deviation was defined as *p* < 0.05.

## 3. Results and Discussion

### 3.1. Changes in Microorganisms

Changes in LAB, yeast, *Staphylococcus*, and Enterobacteriaceae during in vitro human digestion are depicted in Figure 2. Stomach digestion significantly (*p* < 0.05) inhibits the growth of all microorganisms. However, after digestion in the small intestine, these microorganisms proliferated significantly (*p* < 0.05). Before digestion, higher levels of LAB were observed in Lp-120 and MS compared with other samples, suggesting that these starter cultures may promote LAB growth (Figure 2A). This phenomenon was similar to the findings of Liu et al. [27], who observed higher LAB counts in inoculated sausages compared to spontaneously fermented samples. After salivary digestion, the LAB contents increased slightly in all groups, demonstrating that LAB in the samples can continue to grow in the oral environment. In comparison, the LAB counts in all samples were significantly (*p* < 0.05) reduced to 3.60–4.42 log cfu/g after simulated gastric digestion. This changing trend might be attributed to the lower pH of simulated gastric juice (pH < 2.0), which inhibited the growth of LAB in the sample [28]. In addition, relatively higher counts of LAB were still observed in Lp-120 and MS. However, after small intestine digestion, the LAB counts in each group increased significantly (*p* < 0.05) to 7.62–7.99 log cfu/g, which were slightly lower than those in undigested samples. Kim et al. [17] reported a similar result that, although gastric juice digestion would affect the survival of LAB in kimchi, many LAB were still alive after stomach digestion.

Figure 2B shows the yeast contents in all samples ranging from 6.99 log cfu/g to 7.51 log cfu/g, with the sample inoculated with Sc-2018 exhibiting the highest level. There were no significant differences (*p* > 0.05) in the counts of yeast before and after simulated salivary digestion. A significant (*p* < 0.05) reduction in their quantity was observed after stomach digestion (4.16–4.55 log cfu/g), which was mainly related to the low pH environment of gastric juice. During digestion in the small intestine, the level of yeast exhibited a significant (*p* < 0.05) rise and reached the pre-digestion level by the end of digestion. These results demonstrated that small intestine digestion provides favorable conditions for the proliferation of yeast, which can still survive after gastric digestion.

Prior to digestion, the *Staphylococcus* contents in different fermented samples ranged from 4.02 log cfu/g to 4.35 log cfu/g (Figure 2C), with Sx-135 samples having the highest content of *Staphylococcus* among the five groups. Following oral digestion, all groups experienced a significant (*p* < 0.05) decrease in *Staphylococcus* quantities, while no significant (*p* > 0.05) changes were observed in LAB and yeast, suggesting that *Staphylococcus* was more sensitive to environmental changes than LAB and yeast. After exposure to simulated gastric juice, the variation trends of *Staphylococcus* in all samples were similar to those of LAB and yeast, with the *Staphylococcus* contents in all samples decreasing, ranging from 2.70 log cfu/g to 3.20 log cfu/g. Casaburi et al. [29] also confirmed that an acidic environment had obvious effect on inhibiting the growth of *Staphylococcus* in traditional fermented sausages. However, the counts of *Staphylococcus* dramatically (*p* < 0.05) increased to 4.69–5.10 log cfu/g after small intestine digestion, with the final numbers of *Staphylococcus* in each group being higher than those before in vitro human digestion. This trend revealed that simulated small intestine digestion juice is more favorable for the proliferation of *Staphylococcus* compared with fermented fish products.

Enterobacteriaceae contents in the samples are presented in Figure 2D. Before digestion, the counts of Enterobacteriaceae in all samples were 3.33 log cfu/g, 2.85 log cfu/g, 2.93 log cfu/g, 2.97 log cfu/g, and 2.76 log cfu/g, respectively. Additionally, the levels in the inoculated samples were lower than those in spontaneously fermented samples, consistent with the previous finding by Chrun et al. [30] in Cambodian fermented small fish (*Pha-ork kontery*). There were no significant (*p* > 0.05) differences in Enterobacteriaceae contents among all samples after oral digestion. Unexpectedly, no Enterobacteriaceae were detected in all samples after the reaction with simulated gastric juice. This demonstrates that Enterobacteriaceae were completely inhibited during stomach digestion, thus elucidating their greater sensitivity to low pH environments compared to LAB, yeast, and *Staphylococcus* in the samples. Zeng et al. [20] demonstrated a significant (*p* < 0.05) correlation between the number of Enterobacteriaceae and pH, indicating that a low pH environment is a key factor in inhibiting the growth of Enterobacteriaceae. However, the amounts of Enterobacteriaceae in all groups increased, ranging from 2.03 log cfu/g to 2.43 log cfu/g during small intestine digestion, which were lower than those in the undigested samples.

### 3.2. Changes in BAs

BAs, a class of low-molecular-weight basic nitrogen-containing compounds with physiological activities, are widely present in traditional fermented food [31]. Fermented fish contains high levels of free amino acids produced via protein hydrolysis, which can generate various BAs through the action of amino acid decarboxylase [32]. Changes in BAs (biogenic amines) concentrations (PUT, CAD, TYR, SPD, and SPM) according to different starter cultures are illustrated in Figure 3. Before digestion, concentrations of PUT in all samples ranged from 56.28 mg/kg to 129.97 mg/kg, with significant (*p* < 0.05) inhibition observed in Lp-120, Sx-135, and MS (Figure 3A). Subsequently, digestion in the oral phase resulted in a slight increase in NS, Lp-120, Sx-135, and MS. After simulated gastric digestion, spontaneous samples and Sx-135 displayed lower PUT concentrations than those in other groups. During subsequent small intestine digestion, PUT concentrations increased by 95.92%, 26.91%, 42.88%, 21.74%, and 38.41% in all groups, respectively. Furthermore, the final concentrations in all samples were higher than the initial undigested levels. Generally, the variations in PUT amounts in inoculated samples were negligible after oral and gastric digestion. Conversely, PUT concentrations in all samples significantly (*p* < 0.05) increased when treated with simulated small intestine juice. In contrast, inoculated groups showed significantly (*p* < 0.05) lower concentrations compared to the NS, and the decrease in the levels of PUT could indicate that inoculated fermentation has great potential to inhibit the accumulation of PUT during small intestine digestion. 

As shown in Figure 3B, the varying concentrations of CAD in all undigested samples suggest that inoculated fermentation significantly (*p* < 0.05) inhibited CAD in the samples. Following oral digestion, the CAD concentrations remained largely unchanged in all groups, while the CAD contents in NS, Sc-2018, and MS decreased significantly (*p* < 0.05) after gastric digestion, indicating that the low pH environment could reduce CAD accumulation during digestion in the stomach. After small intestine digestion, CAD contents drastically increased (*p* < 0.05) and exceeded those of undigested samples. The NS group exhibited the highest growth rate (133.29%), while the final CAD concentrations in the other groups increased by 52.62%, 102.48%, 110.71%, and 98.72% for Lp-120, Sc-2018, Sx-135, and MS, respectively. These results show that inoculated fermentation may have influenced the reduction in CAD accumulation during small intestine digestion at various degrees, with Lp-120 exhibiting significantly (*p* < 0.05) greater inhibition than the other inoculated groups.

As shown in Figure 3C, TYR concentrations were not obviously altered in all samples until the stomach digestion. In addition, significant (*p* < 0.05) reductions in TYR were detected after the reaction with simulated gastric juice, which may involve changes in pH during the phase of this digestion. During subsequent small intestine digestion, TYR levels in all samples increased significantly (*p* < 0.05), with corresponding increments of 242.68% (NS), 114.23% (Lp-120), 164.18% (Sc-2018), 286.20% (Sx-135), and 227.14% (MS), respectively, which were significantly (*p* < 0.05) higher than those of PUT and CAD. A similar trend was also reported by Kim et al. (2018) in fermented sausages. Furthermore, the increase in TYR in Lp-120 and Sc-2018 was significantly (*p* < 0.05) lower than that in the NS sample. Hence, these two starter cultures are beneficial to suppress the accumulation of TYR during small intestine digestion.

The changes in SPD and SPM contents during in vitro human digestion of CTFPs are shown in Figure 3D,E. There were no obvious differences in SPD and SPM levels after oral digestion. However, digestion with simulated gastric juice significantly (*p* < 0.05) increased the concentrations of SPD and SPM in all samples. The SPD and SPM contents increased by 212.24% to 690.98% and 81.43% to 142.54%, respectively, possibly due to the degradation of PUT and some precursor amino acids. Additionally, the increase in SPD and SPM in Lp-120 and Sc-2018 was lower than that in other groups. During subsequent small intestine digestion, SPD (≤13.74 mg/kg) and SPM (≤25.59 mg/kg) contents rapidly decreased, even lower than the undigested samples. These results indicate that Lp-120 and Sc-2018 starter cultures have a positive effect on reducing the accumulation of PUT and SPD during gastric digestion. Furthermore, the concentration of these BAs decreased to low levels at the end of digestion, which might enhance the food safety of CTFPs.

### 3.3. Changes in Nitrite Contents

The nitrite contents of CTFPs are shown in Figure 4. The initial nitrite levels in CTFPs (<0.2 mg/kg) were determined in our previous work [33]. Lower nitrite contents were observed in Lp-120, Sc-2018, and MS, indicating that these three starter cultures have a notable influence on the accumulation of nitrite during fermentation (*p* < 0.05). After salivary digestion, the nitrite contents did not change significantly (*p* > 0.05) compared to the undigested samples. A similar trend was also reported by Kim et al. [18] for the nitrite concentration of pork patties after oral digestion. During gastric juice digestion, the inhibitory effect of nitrite significantly (*p* < 0.05) decreased in all fermentation groups, with corresponding reductions of 37.33%, 41.05%, 44.13%, 38.91%, and 36.64% in NS, Lp-120, Sc-2018, Sx-135, and MS, respectively, compared to salivary digestion. Our previous study found that the degradation of nitrite was closely correlated with the initial acidic pH conditions [33]. Therefore, such variation existed mainly due to low pH values (pH < 2) in gastric juice. Meanwhile, this corresponds to the aforementioned findings that some LAB and yeast still survived after gastric digestion, which may result in a lower level of nitrite via the enzymatic degradation pathway. Unexpectedly, after small intestine digestion, the nitrite content increased in the samples of NS, Lp-120, Sc-2018, Sx-135, and MS fermentation groups by 18.16%, 14.41%, 17.86%, 6.93%, and 30.73%, respectively. However, the increase in single inoculated groups was lower than that in the natural fermentation group, indicating that inoculating fermentation has the potential to inhibit nitrite increase during small intestine digestion.

### 3.4. Changes in NAs

To investigate whether the autochthonous starter culture inoculation could affect the accumulation of NAs in CTFPs during digestion, the changes in NDMA, NMEA, NPIP, NPYR, NDPheA, and NMOR were assessed. As depicted in Table 2, the levels of NMEA in all samples were under the detection limit, and no statistical differences were (*p* > 0.05) observed from NDPheA during digestion. The concentrations of NDMA in the different fermentation groups before digestion ranged from 1.21 µg/kg to 3.15 µg/kg, and no obvious changes were observed after oral digestion. During subsequent gastric digestion, NDMA levels significantly (*p* < 0.05) increased in all groups compared to those treated with simulated saliva (with increases of 70.09%, 54.48%, 39.37%, 72.48%, and 33.33%, respectively). Williams. [34] also presented that the acidic environment of gastric juice promotes the nitrosation reaction. However, Kim et al. [17] found that no significant (*p* > 0.05) change in NDMA in kimchi was observed during simulated gastric digestion, which may be due to there being little precursor material for NDMA synthesis in kimchi. Whereas the high content of precursors such as TMA, DMA, and nitrite in CTFPs may be related to a significant increase in NDMA during digestion. After gastric digestion, no significant changes (*p* > 0.05) in NDMA concentration were observed. These results indicated that the accumulation of NDMA mainly occurred during digestion in the gastric juice, and Lp-120, Sc-2018, and MS could have influenced the reduction in NDMA during gastric digestion.

The changing trend of NPIP among different samples was similar to that of NDMA. Before digestion, the NPIP content in the control (0.49 μg/kg) was noticeably lower than that in Lp-120, Sx-135, and MS (0.59, 0.56, and 0.51 μg/kg, respectively). Subsequently, there was no change in NPIP accumulation after oral digestion. In contrast, after being treated with simulated gastric juice, the NPIP levels in all groups increased by 66.67%, 67.21%, 67.35%, 93.22%, and 73.08%, respectively, when compared to undigested samples. In addition to nitrosating reagents, CAD and piperidine are considered as precursors of NPIP, the former was mainly from the seasoning added during the processing of meat products [35,36]. However, it was not added in this study, indicating that the accumulation of NPIP during gastric juice digestion may be responsible for the transformation of CAD. After small intestine digestion, there was no significant (*p* < 0.05) change in NPIP concentration, which ranged from 0.80 μg/kg to 1.12 μg/kg in all groups. Apparently, NPIP accumulation in CTFPs mainly occurred during gastric juice digestion, and CTFPs fermented with four different starter cultures had no obvious NPIP depletion ability, as compared to the NS sample. However, the final level of that in all samples after digestion is still acceptable, which may pose no health risks.

The concentrations of NPYR and NMOR did not change obviously during the various stages of digestion, particularly after gastric juice digestion. According to previous observations, some BAs, such as PUT and SPD, might serve as precursors for NPYR, but their nitrosation reactions would not occur without a high-temperature environment (160–170 °C). The in vitro simulated digestion did not provide sufficient temperature conditions to facilitate such reaction [36,37]. Additionally, the precursor substance of NMOR is mainly derived from packaging materials and preservatives [38], which are not present in the CTFPs fermentation systems. The lack of precursor material may be the main reason for the absence of an obvious change in NMOR concentration during digestion. The results also suggest that NPYR and NMOR in CTFPs remained stable during in vitro human digestion.

## 4. Conclusions

This study investigated the impact of distinct starter cultures on the concentration of NAs and their precursors throughout in vitro human digestion. The gastric digestion phase significantly (*p* < 0.05) suppressed the proliferation of LAB, yeast, *Staphylococcus*, and Enterobacteriaceae, while simultaneously reducing the accumulation of nitrite. Subsequently, the microbes and the nitrite contents significantly (*p* < 0.05) increased during intestinal digestion. However, the increased amounts of nitrite in inoculated groups were lower than in the spontaneous fermentation group, indicating that inoculating fermentation had a positive effect on nitrite inhibition during intestinal digestion. Meanwhile, the accumulation of PUT, CAD, and TYR exhibited a notable rise (*p* < 0.05) in the small intestine, while inoculating fermentation retarded the accumulation of these BAs. In addition, significant (*p* < 0.05) enhancement in NDMA occurred after stomach digestion, while LP-120, SC-2018, and mixed starter cultures have reduced the increase in NDMA to various degrees. Therefore, this study was conducted using in vitro digestion as the experimental method, revealing that inoculation with autochthonous starter cultures, particularly LP-120, SC-2018, and mixed starter cultures in CTFPs, demonstrates enormous promise as fermentation starters for enhancing CTFPs’ edible safety during digestion.

## Figures and Tables

**Figure 1 foods-13-02021-f001:**
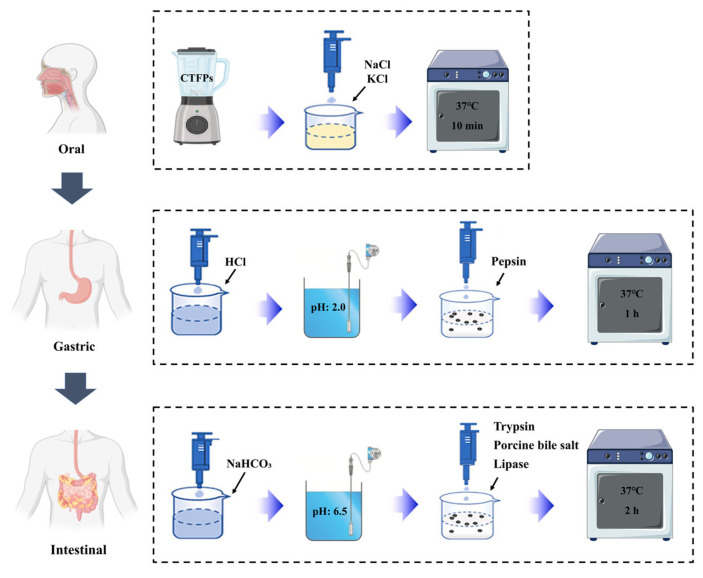
Flow diagram for the in vitro simulated digestion process.

**Figure 2 foods-13-02021-f002:**
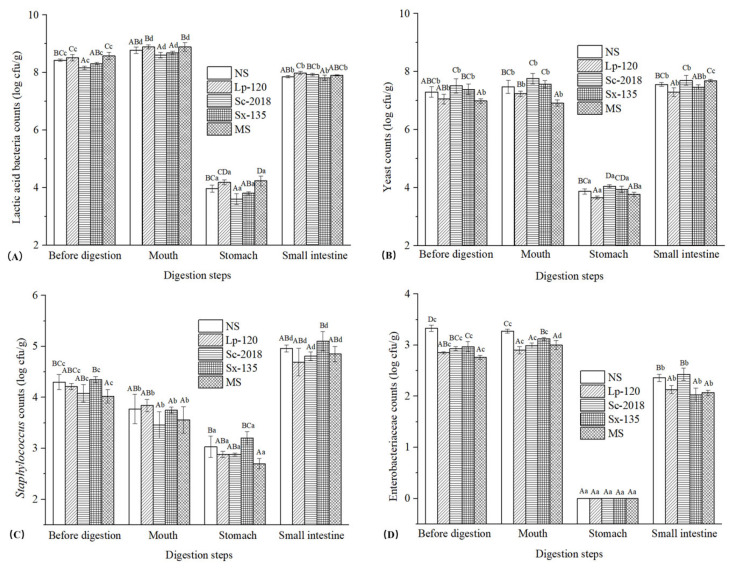
Effects of inoculating *L. plantarum* 120, *S. cerevisiae* 2018, *S. xylosus* 135, or MS on the counts of lactic acid bacteria (**A**), yeast (**B**), *Staphylococcus* (**C**), and Enterobacteriaceae (**D**) in the Chinese traditional fermented fish products during in vitro human digestion. NS: sample without inoculation; MS: sample inoculated with mixed starter cultures (*L. plantarum* 120, *S. cerevisiae* 2018, and *S. xylosus* 135 [1:1:1]). Different capital letters within the same digestion stage indicate significant differences (*p* < 0.05). The different lowercase letters within the same treatment indicate significant differences (*p* < 0.05).

**Figure 3 foods-13-02021-f003:**
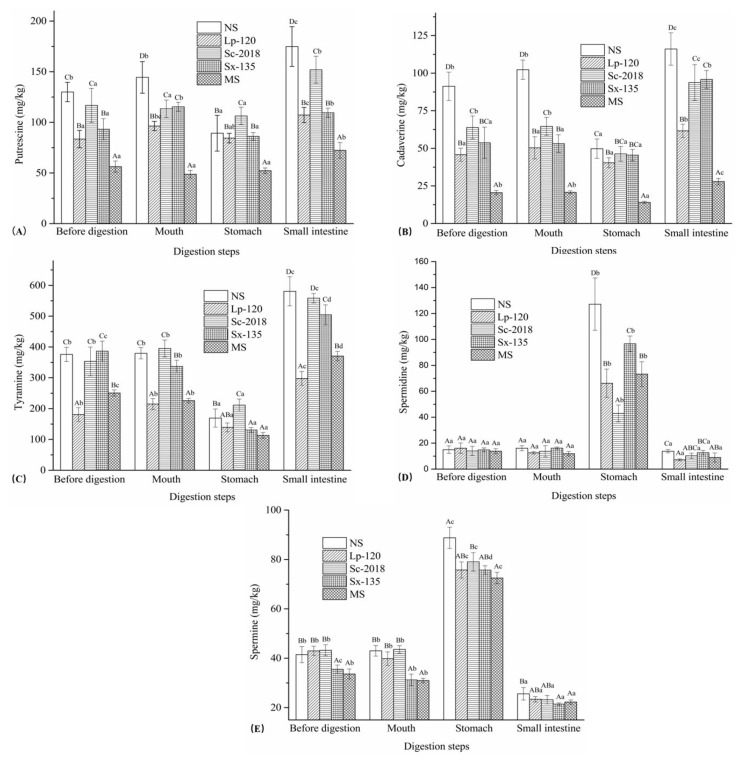
Effects of inoculating *L. plantarum* 120, *S. cerevisiae* 2018, *S. xylosus* 135, or MS on the putrescine (**A**), cadaverine (**B**), tyramine (**C**), spermidine (**D**), and spermine (**E**) contents in the Chinese traditional fermented fish products during in vitro human digestion. NS: sample without inoculation; MS: sample inoculated with mixed starter cultures (*L. plantarum* 120, *S. cerevisiae* 2018, and *S. xylosus* 135 [1:1:1]). Different capital letters within the same digestion stage indicate significant differences (*p* < 0.05). The different lowercase letters within the same treatment indicate significant differences (*p* < 0.05).

**Figure 4 foods-13-02021-f004:**
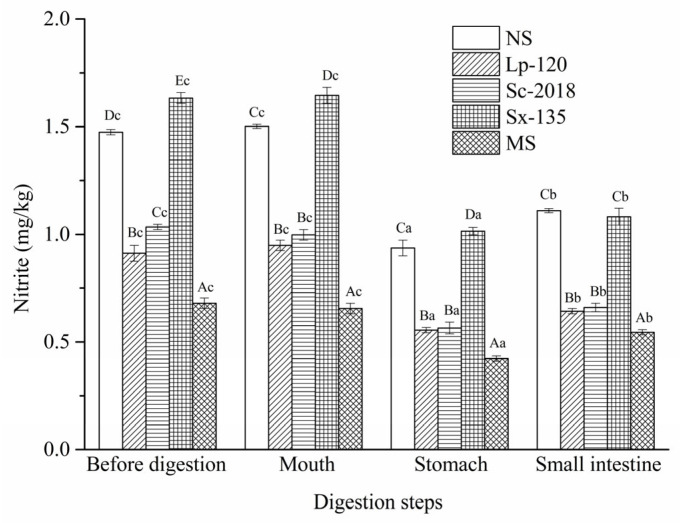
Effects of inoculating *L. plantarum* 120, *S. cerevisiae* 2018, *S. xylosus* 135, or MS on the nitrite contents in the Chinese traditional fermented fish products during in vitro human digestion. NS: sample without inoculation; MS: sample inoculated with mixed starter cultures (*L. plantarum* 120, *S. cerevisiae* 2018, and *S. xylosus* 135 [1:1:1]). Different capital letters within the same digestion stage indicate significant differences (*p* < 0.05). The different lowercase letters within the same treatment indicate significant differences (*p* < 0.05).

**Table 1 foods-13-02021-t001:** Constituents and concentrations of various synthetic solutions used in the in vitro simulated digestion system.

	Saliva (Mouth Step)	Gastric Juice (Stomach Step)	Duodenal Juice (Small Intestine Step)
Organic and inorganic components	10 mL salt solution (140 mmol/L NaCl and 5 mol/L KCl)	5 mL 0.1 mol/L HCl	37.5 mL 0.1 mol/L NaHCO_3_
Enzymes	—	200 mg pepsin	75 mg pancreatin 450 mg porcine bile salt 25 mg lipase
pH	6.8 ± 0.2	2 ± 0.02	6.5 ± 0.2
Conditions	37 °C/200 rmp/10 min	37 °C/200 rmp/1 h	37 °C/200 rmp/2 h

Not added components are shown in “—” (The CTFPs samples contained negligible amounts of starch).

**Table 2 foods-13-02021-t002:** Effects of inoculating *L. plantarum* 120, *S. cerevisiae* 2018, *S. xylosus* 135, or MS on the *N-*nitrosodimethylamine (NDMA), *N-*nitrosoethylmethylamine (NMEA), *N-*nitrosopiperidine (NPIP), *N-*nitrosopyrrolidine (NPYR), *N-*nitrosodiphenylamine (NDPheA), and *N-*nitrosomorpholine (NMOR) contents in the Chinese traditional fermented fish products during in vitro human digestion.

Digestion Steps	Groups	NDMA	NMEA	NPIP	NPYR	NDPheA	NMOR
Before digestion	NS	3.15 ± 0.16 ^Da^	ND	0.49 ± 0.03 ^Aa^	1.15 ± 0.06 ^Da^	0.06 ± 0.01 ^Ba^	0.32 ± 0.02 ^Aa^
	Lp-120	1.54 ± 0.08 ^Ba^	ND	0.59 ± 0.05 ^Ca^	0.74 ± 0.08 ^Ca^	0.05 ± 0.01 ^Ba^	0.34 ± 0.03 ^Aa^
	Sc-2018	2.32 ± 0.07 ^Ca^	ND	0.45 ± 0.02 ^Aa^	1.21 ± 0.05 ^Da^	ND	0.32 ± 0.04 ^Aab^
	Sx-135	3.07 ± 0.03 ^Da^	ND	0.56 ± 0.03 ^BCa^	0.57 ± 0.03 ^Ba^	0.02 ± 0.01 ^Aa^	0.37 ± 0.03 ^ABa^
	MS	1.21 ± 0.02 ^Ab^	ND	0.51 ± 0.02 ^ABa^	0.29 ± 0.04 ^Aa^	ND	0.41 ± 0.02 ^Ba^
Mouth	NS	3.21 ± 0.05 ^Ea^	ND	0.51 ± 0.02 ^Aa^	1.21 ± 0.07 ^Da^	0.06 ± 0.01 ^Ba^	0.31 ± 0.03 ^Aa^
	Lp-120	1.45 ± 0.06 ^Ba^	ND	0.61 ± 0.03 ^Ba^	0.81 ± 0.02 ^Cab^	0.05 ± 0.02 ^Ba^	0.40 ± 0.06 ^Bab^
	Sc-2018	2.21 ± 0.06 ^Ca^	ND	0.49 ± 0.03 ^Aa^	1.26 ± 0.06 ^Da^	ND	0.30 ± 0.02 ^Aa^
	Sx-135	2.98 ± 0.07 ^Da^	ND	0.59 ± 0.04 ^Ba^	0.58 ± 0.02 ^Ba^	0.02 ± 0.01 ^Aa^	0.39 ± 0.02 ^Bab^
	MS	1.05 ± 0.03 ^Aa^	ND	0.52 ± 0.03 ^Aa^	0.35 ± 0.03 ^Aa^	ND	0.35 ± 0.03 ^ABa^
Stomach	NS	5.46 ± 0.21 ^Eb^	ND	0.85 ± 0.02 ^Ab^	1.29 ± 0.11 ^Da^	0.07 ± 0.02 ^Ba^	0.36 ± 0.02 ^Aab^
	Lp-120	2.24 ± 0.11 ^Bb^	ND	1.02 ± 0.06 ^Bb^	0.90 ± 0.11 ^Cb^	0.06 ± 0.01 ^Ba^	0.41 ± 0.03 ^ABab^
	Sc-2018	3.08 ± 0.03 ^Cc^	ND	0.82 ± 0.04 ^Ab^	1.18 ± 0.06 ^Da^	ND	0.38 ± 0.03 ^ABb^
	Sx-135	5.14 ± 0.17 ^Db^	ND	1.14 ± 0.10 ^Cb^	0.54 ± 0.07 ^Ba^	0.03 ± 0.02 ^Aa^	0.44 ± 0.05 ^Bb^
	MS	1.40 ± 0.03 ^Ac^	ND	0.9 ± 0.03 ^Ab^	0.34 ± 0.03 ^Aa^	ND	0.39 ± 0.03 ^ABa^
Small intestine	NS	5.39 ± 0.03 ^Eb^	ND	0.87 ± 0.03 ^Bb^	1.19 ± 0.03 ^Da^	0.06 ± 0.01 ^Ba^	0.39 ± 0.03 ^ABb^
	Lp-120	2.12 ± 0.02 ^Bb^	ND	1.05 ± 0.02 ^Db^	0.85 ± 0.02 ^Cab^	0.05 ± 0.02 ^Ba^	0.42 ± 0.02 ^Bb^
	Sc-2018	2.93 ± 0.05 ^Cb^	ND	0.80 ± 0.05 ^Ab^	1.23 ± 0.05 ^Da^	ND	0.39 ± 0.05 ^ABb^
	Sx-135	5.09 ± 0.02 ^Db^	ND	1.12 ± 0.02 ^Eb^	0.59 ± 0.02 ^Ba^	0.02 ± 0.02 ^Aa^	0.41 ± 0.02 ^ABab^
	MS	1.35 ± 0.04 ^Ac^	ND	0.94 ± 0.04 ^Cb^	0.35 ± 0.04 ^Aa^	ND	0.35 ± 0.04 ^Aa^

NS: sample without inoculation; MS: sample inoculated with mixed starter cultures (*L. plantarum* 120, *S. cerevisiae* 2018, and *S. xylosus* 135 [1:1:1]). Different capital letters within the same digestion stage indicate significant differences (*p* < 0.05). The different lowercase letters within the same treatment indicate significant differences (*p* < 0.05). ND: non-detected value.

## Data Availability

The original contributions presented in the study are included in the article, further inquiries can be directed to the corresponding author.

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
