# Peer review of "Effects of Inoculating Autochthonous Starter Cultures on Changes of N-Nitrosamines and Their Precursors in Chinese Traditional Fermented Fish during In Vitro Human Digestion"

_foods, 2024, doi:10.3390/foods13132021_

Round 1

Reviewer 1 Report

Comments and Suggestions for Authors

The Reviewer’s comments on paper Effects of inoculating autochthonous starter cultures on changes of N-nitrosamines and their precursors in Chinese traditional fermented fish during in vitro human digestion; foods-3028972

The paper covers very topical issue and provides data that will be very helpful in further research of fermented fish products.

However, the authors should make some minor improvements:

 lines 85–87: Cited paper does not provide data about antimicrobial properties of stated strains. Please explain.

 lines 102-103: How was determined ability of MO to degrade BAs. How many mL of inoculum was added in each treatment.

 Please rephrase title of Figures 2-4 and Table 2, they are unclear. e.g. Table 2 Influence of different starter cultures on NAs changes…  

Table 2. Please show data about NDPheA. What is the meaning of letters in brackets in the title of Table 2? Whether instead A after N-nitrosodimethylamine should be NDMA? 

Reviewer 2 Report

Comments and Suggestions for Authors

The manuscript (foods-3028972) deals with an important issue related to the effect of starter cultures on N-nitrosamines and their precursors in Chinese fermented fish during in vitro human digestion. Additionally, microbiological analysis was performed during the in vitro human digestion. Introduction, research methods and description of results are presented in a clear and appropriate manner. However, some first overall recommendations are suggested as follows:

L107-108. Please, provide the fermentation temperature and the package or fermentation conditions. Also, the pH of the fermented fishes should be reported.

Section 2.3. Can you explain how the in vitro human digestion samples for each step were prepared for each analysis (microbiological analysis, BAs determination, nitrites and NAs)? Mixed different batches of 5 g (in vitro human digestion) for each analysis or scaled up the test? For selected test requires 10 g of sample (CTFP), then, how the sample was prepared from the in vitro human digestion? In addition, describe how CTFP sample was removed from solutions, is there any washed o drained step. Please clarify these important steps in the CTFP analysis.

L175-176. Include the range of concentrations used for the standards.

L196. Mention the full name of PSA and C18E the first time.

L308-309, 324-325, 334-335. Could you explain why PUT, CAD and TYR increased during the small intestine digestion?

L344-345. Please, explain more about the PUT and amino acids' degradation to form SPD and SPM.

L375-376. Why nitrites increased during the digestion step? Explain reasonably.

Reviewer 3 Report

Comments and Suggestions for Authors

Comments for the authors:

1.      The introduction lacks a clear hypothesis or research question, which is essential for guiding the reader on what to expect from the study.

2.      Lack of clarity in statistical analysis descriptions – it is unclear which statistical tests were used (215-219).

3.      The choice of inoculating strains (L. plantarum 120, S. cerevisiae 2018, and S. xylosus 135) is not justified in the context of their specific benefits for reducing NAs (lines 84-86).

4.      Missing details on the controls used in the experiments (lines 99-112).

5.      The results/conclusions sections demonstrate some overgeneralization of results, e.g. lines 282-283, inference on pH effects on bacteria without direct data shown;  lines 422-426 fail to address conflicting results such as the stability of NPYR and NMOR during digestion; line 453-454 future research directions are suggested without clear connections to the study’s findings.

Please take the questions:

1.      Could you please specify whether the nitrites measured in the study were introduced as part of the fermentation process through additives, or were they present as a result of initial contamination or natural processes within the ingredients used?

2.      Could you please provide information on the initial nitrite levels in the fish used for fermentation before any treatment? Understanding the baseline nitrite levels would be crucial for a comprehensive assessment of the fermentation process's effectiveness as detailed in your study. This data would greatly aid in comparing the impact of different starter cultures on nitrite reduction. Now it seems that you failed to address some conflicting results found in the study (e.g., nitrite inhibition vs. increase).

3.      Given the crucial role of protein content in the formation of biogenic amines, could you provide more details on the initial protein levels in the fish used for fermentation? Understanding this could significantly deepen the analysis of biogenic amine production during the fermentation process facilitated by different starter cultures.

4.      To gain a more comprehensive understanding of the factors influencing biogenic amine production in your study, could you provide detailed information on the initial microbial load of the fish samples used for fermentation? Specifically, were there baseline assessments of bacterial types and concentrations before the application of the starter cultures? Information on any pre-fermentation treatments that might have affected the microbial profile would also be highly beneficial.

5.      Could you provide information regarding the initial levels of biogenic amines in the Chinese traditional fermented fish products before the fermentation process began?

Round 2

Reviewer 2 Report

Comments and Suggestions for Authors

The authors attended most suggestions and completed the missing information. The manuscript was improved properly.